# Exercise Training in Post-COVID-19 Patients: The Need for a Multifactorial Protocol for a Multifactorial Pathophysiology

**DOI:** 10.3390/jcm11082228

**Published:** 2022-04-15

**Authors:** Gaia Cattadori, Silvia Di Marco, Massimo Baravelli, Anna Picozzi, Giuseppe Ambrosio

**Affiliations:** 1IRCCS Multimedica, 20138 Milan, Italy; silvia.dimarco@multimedica.it (S.D.M.); massimo.baravelli@multimedica.it (M.B.); anna.picozzi@multimedica.it (A.P.); 2Division of Cardiology, University of Perugia School of Medicine, 06123 Perugia, Italy; giuseppe.ambrosio@ospedale.perugia.it

**Keywords:** exercise, training, COVID-19

## Abstract

The battle against COVID-19 has entered a new phase with Rehabilitation Centres being among the major players, because the medical outcome of COVID-19 patients does not end with the control of pulmonary inflammation marked by a negative virology test, as many patients continue to suffer from long-COVID-19 syndrome. Exercise training is known to be highly valuable in patients with cardiac or lung disease, and it exerts beneficial effects on the immune system and inflammation. We therefore reviewed past and recent papers about exercise training, considering the multifactorial features characterizing post-COVID-19 patients’ clinical conditions. Consequently, we conceived a proposal for a post-COVID-19 patient exercise protocol as a combination of multiple recommended exercise training regimens. Specifically, we built pre-evaluation and exercise training for post-COVID-19 patients taking advantage of the various programs of exercise already validated for diseases that may share pathophysiological and clinical characteristics with long-COVID-19.

## 1. Introduction

The coronavirus-2 (SARS-CoV-2) is a new disease that is causing a respiratory illness outbreak (COVID-19). It was first identified in December 2019 in China (Wuhan), subsequently spreading throughout the world and becoming a worldwide pandemic [1].

COVID-19 can be described as a multisystem disease with acute and, likely, chronic consequences, as the grim outcome of COVID-19 survivors does not end with the end of pulmonary inflammation. Data from the UK’s Office for National Statistics suggest a prevalence of post-COVID-19 syndrome or long-COVID of about 13.7%, making crucial the need for rehabilitation interventions to promote physical recovery [2]. Consequently, our battle against COVID-19 has entered a new phase that sees Rehabilitation Centres as major players due to the COVID-19 survivors’ sequelae.

## 2. COVID-19

Is it a chronic disease? At the end of the viral and inflammation phase causing the active disease, most patient are discharged without breathlessness at rest, yet often with poor exercise tolerance associated with persistency of COVID-19 signs at RX or CT pre-discharge evaluation (i.e., long-COVID-19 syndrome). Elevated levels of inflammatory cytokines could persist at follow up, causing vascular remodelling and endothelial dysfunction, possibly leading to pulmonary hypertension [3].Is it a multifactorial disease? To date, there is paucity of data about the precise mechanisms underpinning COVID-19 and no single interpretation may unify the pathophysiological mechanisms underlying the disease and its consequences, which conceivably are multifactorial. Alterations associated with COVID-19, especially in patients requiring ICU care, involve respiratory function (impairment of alveolar air exchange, decrease in pulmonary ventilation, respiratory muscle dysfunction and, probably, pulmonary fibrosis in the long run), cardiac function (reduced systolic function in some cases and possible persistent myocardial damage in the long run), pulmonary vessels (pulmonary hypertension in some cases due to pulmonary embolism and/or thrombosis), peripheral muscle function (due to deconditioning and decreased lean body mass, fatigue and the effects of hypokalaemia) [4,5], and, likely, liver, kidney, and brain and nervous and immune systems [6]. Finally, decreased exercise capacity is the most common dysfunction (61,4% of discharged mild patients) mainly due to the long-term immobilization or to the muscle invasion by the virus [7].

## 3. Inflammation and Exercise Training

Exercise training is known to positively affect immune system and inflammation [3]. The acute inflammatory response may be reduced by a regular physical activity through at least five mechanisms: (1) reducing the inflammatory signalling pathway mediated by Toll-like receptors; (2) increasing anti-inflammatory cytokines such as Interleukin-10 and 37, which could inhibit the inflammatory cascade; (3) reducing lung inflammation promoting the conversion from Angiotensin II to Angiotensin 1–7; (4) activating the Angiotensin-converting enzyme 2 receptor vasodilator pathway to reduce lung inflammation; and (5) restoring nitric oxide levels in order to counteract endothelial dysfunction [8]. However, different physical activities in terms of intensity and type have different effects on the immune system and inflammation: intense exercise can actually lead to a higher level of inflammatory mediators and to increase the risk of injury and chronic inflammation, while moderate-to-vigorous effort with appropriate resting periods can achieve maximum benefit [9]. The “J curve” concept hypothesizes that excessive bouts of prolonged training can impair immune function, and high intensity exercise may thus be dangerous, helping to exacerbate virus infection, such as COVID-19. On the contrary, moderate intensity exercise improves the immune system and it should be recommended as a non-pharmacological, inexpensive and viable way to cope with COVID-19 virus. The “Forrest Gump” theory states, based on study on ACE axis unbalance, that “regular exercise would not reduce one’s risk of getting infected with SARS-CoV-2 but it would reduce one’s risk of getting severe disease [10,11]”. Moreover, several studies have demonstrated that both acute and chronic exercise training at moderate intensity, improve endothelial dysfunction, muscular blood supply, peripheral O2 extraction, muscular strength, ventilator efficacy, resulting in clinically significant benefits in terms of improved exercise capacity, quality of life and cardio-pulmonary function. Exercise programs in adults hospitalized with an acute or an exacerbation of a chronic respiratory condition, even if disparate, were well tolerated, and adverse events were infrequent with movement out of bed within 24 h of hospitalization with progressive daily movement and progression titrated based on symptoms [12].

## 4. Exercise Training in Post-COVID-19 Patients

Exercise training is an integral component of evidence-based management programs for many chronic conditions, particularly those involving cardiac and/or pneumological conditions. Consequently, it would appear logical to extrapolate the exercise training scheme already applied to other chronic conditions to long-COVID-19 patients. General recommendations of the European Society of Cardiology advise “to be prepared to handle COVID-19 patients” [13,14], but among Expert Consensus publications about rehabilitation in COVID-19 patients, only a few papers have evaluated the exercise prescriptions in detail, leading to very generic final suggestions [4,15,16,17,18]. Similarly, only a few randomized clinical trials have been performed regarding the safety and efficacy of different exercise programs in COVID-19 patients, with too few patients enrolled to allow evidence-based recommendations. Specifically, Chen et al. [19] published a systemic review and meta-analysis about the effect of pulmonary rehabilitation for patients with post-COVID-19, including 3 studies with 233 patients. Tested treatment regimens were device-based respiratory training, cough exercise, diaphragmatic training and stretching exercise. Data showed that pulmonary rehabilitation significantly improved the exercise capacity. More recently, McNarry et al. [20] enrolled 281 COVID-19 patients in a randomized controlled trial, demonstrating that inspiratory muscle training improved symptoms, respiratory muscle strength and aerobic fitness. Ahmadi Hekmatikar et al. [21] published a systematic review about functional and psychological changes after exercise training in post-COVID-19 patients, including 7 studies with 286 patients. They showed that training programs composed of aerobic and resistance exercise may improve the functional capacity and quality of life, but a meta-analysis was not conducted because the included studies had methodological heterogeneities and they did not examine a control group. Even though conclusive validations are scarce with the need for future testing in randomized controlled trials and in real life, we tried to build a scheme of exercise training based on the available data for COVID-19 patients at the moment (Table 1).

Consequently, in addressing the urgent need for a structured exercise program for long-COVID-19, it being considered as a multifactorial disease, we reviewed exercise training recommendations validated for similar diseases from a pathophysiological point of view [22,23,24,25,26,27,28,29,30,31,32,33,34,35,36,37,38,39]. Table 2 reports different pathophysiology features of COVID-19 and related landmark diseases with specific characteristics.

We summarized review data about exercise prescription in Table 3, reporting different exercise training programs with a general description, COVID-19 related diseases trial data, and COVID-19 trial or Expert Consensus data.

Based on our review, we conceived a proposal of pre-evaluation (Table 4) and exercise training (Table 5) in post-COVID-19 patients as a mixture of different validated programs of linked diseases from a pathophysiological point of view [32,33,34,35,36,37,38,39] (Figure 1).

## 5. The “New Combined Post-COVID-19 Exercise Protocol”

Preliminary evaluation. Accurate global assessment of post-COVID-19 patients before training is a crucial point to tailor the exercise protocol. Table 4 reported the recommended scales and tests in the previous published COVID-19 trials and COVID-19 pertinent disease protocols [15,16,17,18,19,20,21,22,23,24,25,26,27,28,29,30,31,32,33,34,35,36,37,38,39,40]. We highlighted in **bold type** the most popular and easy to perform ones as the minimum required to set the exercise program.

Patients’ selection. The “new combined post-COVID-19 exercise protocol” has been designed for patients with confirmed diagnosis of COVID-19. A safe start for an exercise protocol is suggested 2 weeks after the cessation of severe symptoms and 1 week from mild/moderate COVID-19 illness [4]. Exclusion criteria are: clinical instability, such as heart rate > 100 bpm, blood pressure < 90/60 mmHg or > 140/90 mmHg, blood saturation < 95%, temperature fluctuation, exacerbation of respiratory symptoms or fatigue not alleviated with rest; other disease that are not suitable for exercise; post-intensive care syndrome, posterior reversible encephalopathy syndrome, critical illness myopathy/neuropathy, neurological or neuro-muscular illness; post COVID-19 myocarditis. Specifically, we focused on cardiorespiratory training of COVID-19 patients, excluding non-stable situation and patients affected by myocarditis for which exercise restriction is mandatory until normalization of ventricular function and absence of inflammation biomarkers and inducible arrhythmias (usually for 3–6 months) [54]. Moreover, clinical presentation/complications such as post-intensive care syndrome, posterior reversible encephalopathy syndrome, critical illness myopathy/neuropathy, neurological or neuro-muscular illness, cognitive deficit and psychological sequelae are out of the topic of the present paper, regarding, specifically, the neuro-COVID-19 unit.

Exercise protocol. General suggestions on exercise training [55] specified a multiple exercise program composed of aerobic exercise (200–400 min per week for 5–7 days per week) and resistance training (two sessions per week). Early rehabilitation seems not well tolerated with rapid desaturation. A scheme of 3 weeks ICU followed by 3 weeks acute medical ward and 3 weeks inpatient rehabilitation should be a good option [56]. More recently, the Stanford Hall Consensus Statement [4] recommended to avoid exercise (>3 METS) for between 2 and 3 weeks after the cessation of severe symptoms and 1 week from mild/moderate COVID-19 illness. The proposed exercise protocol is 12 weeks long as a standard suggestion, and it should be carried out in a Rehabilitation Centre under a specialist supervision for safety reasons for at least 2 weeks; at the end of the 2 weeks, the patients can carry it out independently in their own homes or continue in in-hospital setting according to clinical condition and/or patient’s preference.

Table 5 reported in details the “new combined post-COVID-19 exercise protocol”.

Final evaluation. It should be interesting to repeat preliminary evaluation (Table 4) after at least 12 weeks to establish the effectiveness of the “new combined post-COVID-19 exercise protocol” in terms of exercise capacity, quality of life and cardio-pulmonary function.

## 6. Conclusions

COVID-19 is a multisystem disease with acute and, quite often, chronic consequences, even though limited data are available for exercise prescription in long-COVID-19 patients. The sequelae in those who survive this illness will potentially dominate medical practice for years and rehabilitation medicine should be at the forefront of guiding care for the affected population. We reviewed the previously published protocols on exercise training to build a “new combined post-COVID-19 exercise protocol” tailored for post-COVID-19 patients conceived as frail subjects with interstitial lung disease, likely complicated by cardiac and vascular diseases, as assessed by a specific preliminary evaluation. Future studies are needed to confirm the safety and the efficacy of the “new combined COVID-19 exercise protocol” as a promising strategy to manage long-COVID-19 patients.

## Figures and Tables

**Figure 1 jcm-11-02228-f001:**
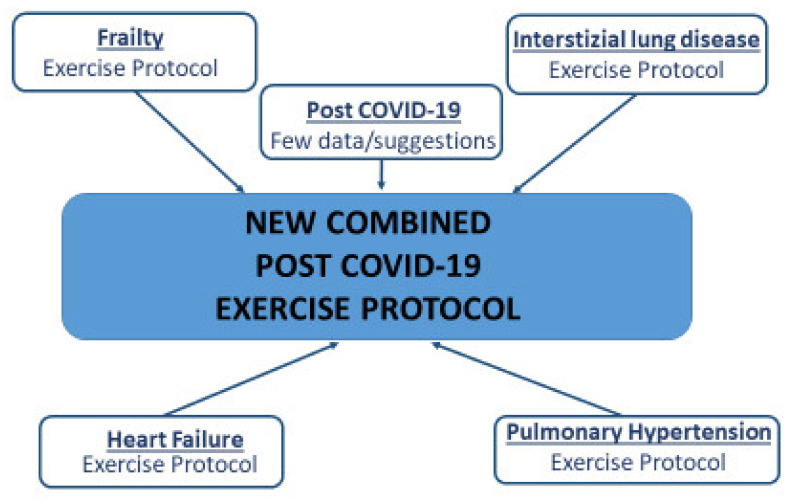
The “new combined post COVID-19 exercise protocol” construction scheme.

**Table 1 jcm-11-02228-t001:** COVID-19 training [15,16,17,18,19,20,21] based on the few available data.

COVID-19 Patients Training
- 4/5 sessions/week for 6 weeks - **Aerobic training:** cycle ergometer, steps, walking, treadmill running; 5–30′ at 40–60% max heart rate or 4–6 Borg Scale - **Resistance training:** upper and lower body exercises; 30–80% of 1 RM; 8–12 repetitions - **Respiratory training:** using a commercial hand-held resistance for 3 sets with 10 breaths 2 times/day for 4 weeks in each set at 60% of maximal expiratory mouth pressure with a rest period of 1 min - **Cough exercise:** 3 sets of 10 active coughs - **Diaphragmatic muscle training:** 30 maximal voluntary diaphragmatic contractions in the supine position, placing a medium weight (1–3 kg) on the anterior abdominal wall to resist diaphragmatic descent - **Stretching exercise:** the respiratory muscles are stretched under the guidance of a rehabilitation therapist; the patient was placed in the supine or lateral decubitus position with the knees bent to correct the lumbar curve; patients were ordered to move their arms in flexion, horizontal extension, abduction and external rotation

RM: one repetition maximum, such as maximal weight an individual can lift for just one repetition with correct technique is the gold standard for assessing strength.

**Table 2 jcm-11-02228-t002:** Different pathophysiology features of COVID-19 and related pertinent diseases.

COVID-19 Pathophysiology Features [4,5,6,7,40]	Related Landmark Diseases [22,23,24,25,26,27,28,29,30,31,32,33,34,35,36,37,38,39]
Respiratory distress with impairment of alveolar air exchange, decrease in pulmonary ventilation and, probably, pulmonary fibrosis in the long run	SARS
	Interstitial lung disease
	Idiopathic pulmonary fibrosis
Pulmonary vessels dysfunction with pulmonary hypertension in some cases due to pulmonary embolism and/or thrombosis	Pulmonary Hypertension
	Interstitial lung disease
	Idiopathic pulmonary fibrosis
Decreased exercise capacity and musculoskeletal deterioration due to the long-term immobilization or to the muscle invasion by the virus, leading to a “frail” post-COVID-19 population	Frailty
Symptomatic high heart rate	Heart Failure
	Interstitial lung disease
	Idiopathic pulmonary fibrosis
	Pulmonary Hypertension
Cardiac dysfunction: reduced systolic function in some cases and possible persistent myocardial damage in the long run	Heart Failure

**Table 3 jcm-11-02228-t003:** Different exercise training programs with a general description, COVID-19 related diseases trial data and COVID-19 trial/Expert Consensus data.

Training	General Description	COVID-19 Related Diseases Trial Data	COVID-19 Trial/Expert Consensus Data
**Continuous** **Aerobic** **training**	Characterized by continuous, dynamic, rhythmic activities involving major muscle groups (i.e., walking, treadmill, cycle ergometer, stair climbing, rower, elliptical trainers) Typically performed at submaximal intensity with the main purpose of progressively moving the anaerobic threshold Heart rate or oxygen consumption measurement to set training intensity.	SARS-CoV-1	COVID-19 trial Expert Consensus
		Frailty	
		Interstitial lung disease	
		Idiopathic pulmonary fibrosis	
		Heart Failure Pulmonary Hypertension	
**Interval** **Training**	High/Low intensity: intermittent periods of high/low intensity exercise separated by periods of low intensity/recovery Heart rate or oxygen consumption measurement to set training intensity.	Heart Failure	COVID-19 trial
**Resistance** **Training**	Primarily anaerobic physical exercises designed to promote muscles force against external weights. 1RM (one-repetition maximum), the maximum amount of weight that a person can possibly lift for one repetition, used to set training intensity. It promotes less pronounced cardiorespiratory responses when compared to aerobic exercise	SARS-CoV-1	COVID-19 trial Expert Consensus
		Frailty	
		Interstitial lung disease	
		Idiopathic pulmonary fibrosis	
		Heart Failure	
		Pulmonary Hypertension	
**Inspiratory** **muscles** **training**	Inspiration using a commercial hand-held resistance	SARS-CoV-1	COVID-19 trial Expert Consensus
		Heart Failure	
		Pulmonary Hypertension	
**Cough** **Exercise**	Sets of active cough under the guidance of a rehabilitation therapist		COVID-19 trial Expert Consensus
**Diaphragm** **Training**	Maximal voluntary diaphragmatic contractions in the supine position, placing a medium weight (1–3 kg) on the anterior abdominal wall to resist diaphragmatic descent		COVID-19 trial Expert Consensus
**Stretching** **Exercise**	The respiratory muscles stretched under the guidance of a rehabilitation therapist; the patient placed in the supine or lateral decubitus position with the knees bent to correct the lumbar curve; patients ordered to move their arms in flexion, horizontal extension, abduction and external	Idiopathic pulmonary fibrosis	COVID-19 trials Expert Consensus
**Flexibility** **Exercise**	Static and dynamic stretching leading to progressive increase in range of motion	Frailty	
		Idiopathic pulmonary fibrosis	
**Balance** **Exercise**	Leg stances, semi-tandem and tandem stance, toe walking, heel walking, tandem gait, walking on a balance board, eye–hand and eye–leg coordination	Frailty	Expert Consensus
**Deep/** **slow breath sessions**	Special form of training skilfully mastered by patients through a series of choreographed action routines and with the help of words, pictures, videos or other communication methods. During breathing training, it is necessary to pay attention to the coordination of diaphragm movement with trunk and limb movement so that diaphragm-function training, breathing-mode training and body and joint training can be carried out at the same time.	Idiopathic pulmonary fibrosis Heart Failure Pulmonary Hypertension	Expert Consensus

**Table 4 jcm-11-02228-t004:** Preliminary evaluation [15,16,17,18,19,20,21,22,23,24,25,26,27,28,29,30,31,32,33,34,35,36,37,38,39,40].

Evaluation	Scales or Tests
**Disability**	BARTHEL Index [41] Activities of Daily Living scale (ADL) [42] **Short Physical Performance Battery (SPPB)** [43]
**Frailty**	Fried’s Frailty Phenotype [44] Frailty Index of Accumulative Deficits [45] 5 m Gait Speed [46] SPPB [43]
**Strength**	**Hand Grip Test** [47]
**Endurance**	Cardiopulmonary Exercise Test (if available) [48] **6 Minute Walking Test (SpO2 + respiratory rate + Borg scale before/after)** [48]
**Balance**	Berg Balance Scale [49] SPPB [43]
**Respiratory Function**	Rest/nocturnal SpO2 [32] **Spirometry** [32] Diffusion capacity [32] **Maximal Inspiratory/Expiratory Pressure (MIP/MEP)** [29]
**Cardiovascular Function**	**Transthoracic echocardiogram** [48]
**Questionnaire**	International Physical Activity Questionnaire-Short Form (IPAQ-SF) [50] Physical Activity Scale for the Elderly (PASE) [51] Kansas City Cardiomyopathy Questionnaire (KCCQ) [52] St. George’s Respiratory Questionnaire (SGRQ) [53]

**Table 5 jcm-11-02228-t005:** The new combined post-COVID-19 exercise protocol.

Training	Modality	Frequency	Intensity	Duration
**Aerobic** **continuous training**	Walking or cycling	2→5 days/week; 150–300 min/week	Walking 80% of peak walking speed achieved on the 6 MWt; Cycling at 50–60%→70% WR max or 60–75%-->80–85% HR max estimated from 6 MWt or Borg 4–6→10; between AT and RC estimated from CPET	20–30 min→65 min per session; 8–12 weeks long
**Interval** **training**	Walking or cycling		Short bouts (10–30 s) of moderate–high intensity at 50–100% peak exercise capacity and a longer recovery (80–60 s)	30 min aerobic interval training (5 min bout + 1 min rest repeated 5 times)
**Resistance/** **Strength** **training**	Upper and lower body strength	2–3→5 times/week	10–15→40→80% of 1 RM; 3–5 on Borg scale; wall push-ups, chair squat, dumbbells shoulder press, dumbbells biceps curls, dumbbells arm extension and abdominal curl-ups	8–12→15 repetitions with 1 min of rest between steps for 1–3→4–6 times; 10→45 min for each session
**Inspiratory** **muscles** **training**	Using a commercial hand-held resistance	2 times/day; 2 sessions/week	60% of maximal expiratory mouth pressure	3 sets with 10 breaths in each set with a rest period of 1 min
**Cough** **Exercise**	Under the guidance of a rehabilitation therapist	2 sessions/week		3 sets of 10 active coughs
**Diaphragmatic muscle** **training**	Supine position	2 sessions/week	placing a medium weight (1–3 kg) on the anterior abdominal wall to resist diaphragmatic descent	30 maximal voluntary diaphragmatic contractions
**Stretching** **Exercise**	Supine or lateral decubitus with the knees bent to correct the lumbar curve, moving their arms in flexion, horizontal extension, abduction and external rotation		Titrate to symptoms	One set of 4–5 stretching exercises for 15–30 s
**Balance** **Exercise**	Leg stances, semi-tandem and tandem stance, toe walking, heel walking, tandem gait, walking on a balance board, eye–hand and eye–leg coordination; under the guidance of a rehabilitation therapist	2–3 days/week		Among the different training days
**Flexibility** **Exercise**	Static and dynamic stretching leading to progressive increase in range of motion; dynamic stretching in warm-up, whereas static stretching exercise at the end in the cool-down phase; under the guidance of a rehabilitation therapist	2–3→5 days/week	5 min long	
**Slow breathing Sessions**	The patient connected to a device providing rhythmic sounds for the progressive lowering of the respiratory rate	6 b/min 30′ daily		20–30 min for every daily session

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
