# Peer review of "Exercise Training in Post-COVID-19 Patients: The Need for a Multifactorial Protocol for a Multifactorial Pathophysiology"

_jcm, 2022, doi:10.3390/jcm11082228_

Round 1

Reviewer 1 Report

The present paper is in line with the current situation to talk about COVID-19 in response to the current outbreak. However, the prevention of the emergence and spread of disease is more important than the prognosis and recovery of critically ill patients. It may be more practical to propose specific exercise prescriptions for THE prevention of COVID-19.

The description of the background part is too simple, and the specific plan is not elaborated in detail. However, 12 references on exercise prescription of COVID-19 are simply referred to, and other guidance opinions are not extensively consulted and used for reference, so there is no evaluation standard.

The exercise program is implemented for only one patient, and its general acceptance, practicability, popularity and feasibility still need to be further discussed.

The review section is too thin. It does not comprehensively summarize the advantages and disadvantages of various exercise programs and the similarities and differences. It does not discuss specific training means and guide patients how to carry out training according to the specific situation, symptoms, disease degree and other factors.

Lack of comparison between aerobic exercise, resistance exercise, interval exercise and respiratory training in the treatment of COVID-19.

It is suggested to summarize different exercise prescriptions for different sequelae of COVID-19 and provide more evidents on its pathophysiology.

Author Response

Response to Reviewer 1.

We want to take the opportunity to thank the reviewer for their insightful comments. We believe that the suggestions have led to substantial improvement of our manuscript. Below we address all questions and provide our responses to the comments.

The present paper is in line with the current situation to talk about COVID-19 in response to the current outbreak. However, the prevention of the emergence and spread of disease is more important than the prognosis and recovery of critically ill patients. It may be more practical to propose specific exercise prescriptions for the prevention of COVID-19.

It is absolutely true that the prevention of the emergence and spread of COVID-19 is very important. However, worldwide epidemiology in those past 2 years have shown that  it is quite hard to predict who, and when, will eventually become infected. And therefore it would be difficult to design an exercise protocol aimed at preventing COVID-19.

On the other hand, COVID-19 patients may experience unexplained and persisting signs or symptoms over 12 weeks and residual impairments in physical function 1 to 2 years after infection. “Data from the UK’s Office for National Statistics suggest a prevalence of post-COVID-19 syndrome or long COVID of about 13,7%”, making crucial the need of rehabilitation interventions to promote physical recovery”. This was the reasoning behind our proposal.

The description of the background part is too simple, and the specific plan is not elaborated in detail. However, 12 references on exercise prescription of COVID-19 are simply referred to, and other guidance opinions are not extensively consulted and used for reference, so there is no evaluation standard.
We thank the reviewer for this important suggestion.

We improved the description of the background reporting extensively main trials and guidance opinions about COVID-19 patients, including new papers published more recently during the revision period.

Specifically, the text now reads “among Expert Consensus publications about rehabilitation in COVID-19 patients, only few papers evaluated the exercise prescriptions in detail leading to very generic final suggestions. Similarly, only few randomized clinical trials were performed about safety and efficacy of different exercise programs in COVID-19 patients with too little patients enrolled to allow evidence-based recommendations. Specifically, Chen et al. published a systemic review and meta-analysis about the effect of pulmonary rehabilitation for patients with post-COVID-19, including 3 studies with 233 patients. Tested treatment regimens were devise-based respiratory training, cough exercise, diaphragmatic training and stretching exercise. Data showed that pulmonary rehabilitation significantly improved the exercise capacity. More recently, McNarry et al. enrolled 281 COVID-19 patients in a RCT demonstrating that inspiratory muscle training improved symptoms, respiratory muscle strength and aerobic fitness. Ahmadi Hekmatikar et al published a systematic review about functional and psychological changes after exercise training in post-COVID-19 patients, including 7 studies with 286 patients. Evidence showed that training programs composed of aerobic and resistance exercise may improve the functional capacity and quality of life, but a meta-analysis was not conducted because the included studies had methodological heterogeneities and they did not examine a control group”. We changed table 1 and text accordingly.      

The exercise program is implemented for only one patient, and its general acceptance, practicability, popularity and feasibility still need to be further discussed.

We thank again the reviewer for the comment about the absence of exhaustive discussion around exercise program, mainly due to the paucity of available data about exercise training in long COVID-19 patients. We underlined this crucial point in the text and “even if conclusive validations are scarce with the need of future testing in randomized clinical trial and in real life, in Table 1 we tried to build a scheme of exercise training based on the available data for COVID-19 patients at the moment”. Moreover, in the conclusion section we wrote “future studies are needed to confirm the safety and the efficacy of the new combined COVID-19 exercise protocol as a promising strategy to manage long COVID-19 patients”.

The review section is too thin. It does not comprehensively summarize the advantages and disadvantages of various exercise programs and the similarities and differences. It does not discuss specific training means and guide patients how to carry out training according to the specific situation, symptoms, disease degree and other factors. Lack of comparison between aerobic exercise, resistance exercise, interval exercise and respiratory training in the treatment of COVID-19. It is suggested to summarize different exercise prescriptions for different sequelae of COVID-19 and provide more evidents on its pathophysiology.

We thank again the reviewer for the important suggestions.

We built table 2 reporting COVID-19 different pathophysiology and related landmark diseases for specific characteristic.

Moreover, we built table 3 reporting different exercise programs with general description, COVID-19 related diseases validation data and specific COVID-19 data.

We consequently deleted table 2-6 making the paper much more easy reading.

We changed the text accordingly.

“Consequently, solving the urgent need of a structured exercise program for long COVID-19 considered as a multifactorial disease, we reviewed exercise training recommendations validated for similar disease from a pathophysiological point of view. Table 2 reported COVID-19 different pathophysiology and related pertinent disease.”

 ”Finally, we summarized review data in table 3 reporting different exercise training programs with general description, COVID-19 related diseases validation data and specific COVID-19 data”.

Reviewer 2 Report

Silvia Di Marco et al. aim in their study to review the previous exercise protocols designed for post-covid patients, people with various lung diseases and heart failure patients, and to create a new exercise protocol for post covid patients. The topic is interesting and clinically relevant. However, the study has some serious and major limitations.

  1. The exercise protocol provided by the authors is too broad and aims to cover several groups of patients. Please focus only on post-covid patients without any severe comorbidities.
  2. Several methods are listed for preliminary evaluation. Should clinican perform all of these tests for every patients (not realistic)? Does the choice of test depend on the patient's complaints? Please make it clear which evaluation methods are required for all patients, which are alternatives of each other, and which are optional, according to your recommendations.
  3. It is not clear whether the proposed exercise protocol should be carried out in a rehabilitation centre under the supervision of a specialist or whether patients can carry it out independently in their own homes.
  4. The protocol lists several forms of training. Do you recommend that patients should do all of them, or is there any criterion system for choosing which form of exercise is recommended for whom? Please clarify.
  5. Tables are difficult to read and the abbreviations used are inconsistent. Please standardise the abbreviations.
  6. Traditional Chinese medicine respiratory rehabilitation is included in the protocol but there is no description or reference provided, which would be necessary, as this method is not widely known.
  7. The leading symptom in many post-covid patients is a high heart rate at rest or on exercise. This problem is not mentioned in the article. What rehabilitation training is recommended for these patients?

Author Response

Response to Reviewer 2.

We want to take this the opportunity to thank you the reviewer for considering our manuscript and also mark our appreciation for your insightful comments. We believe that your the suggestions have led to substantial improvement of our manuscript. Below we address all the questions and provide our responses to your the comments.

1.The exercise protocol provided by the authors is too broad and aims to cover several groups of patients. Please focus only on post-covid patients
We thank the reviewer for the suggestion.

We simplify the exercise protocol focusing only on post-COVID-19 patients.

We changed Figure 1 and text accordingly.

“Based on our review data, we conceived a proposal of pre-evaluation and exercise training in post COVID-19 patients as a mixture of different validated programs of related diseases from a pathophysiological point of view.”.

2. Several methods are listed for preliminary evaluation. Should clinican perform all of these tests for every patients (not realistic)? Does the choice of test depend on the patient's complaints? Please make it clear which evaluation methods are required for all patients, which are alternatives of each other, and which are optional, according to your recommendations.

We thank again the reviewer for the suggestion.

The reported evaluation methods are validated in randomized clinical trial reviewed but, we agree with the reviewer that it is not realistic to force the entire list. Therefore, we just highlighted scales/tests as the most popular and easy to perform ones as the minimum required to set the exercise program”.

We changed the table and the text accordingly.

3. It is not clear whether the proposed exercise protocol should be carried out in a rehabilitation centre under the supervision of a specialist or whether patients can carry it out independently in their own homes.

We complete the text to clarify the protocol setting.

“The proposed exercise protocol is 12 weeks long as standard suggestion and it should be carried out in a Rehabilitation Centre under a specialist supervision for at least 2 weeks for safety reasons; at the end of the 2 weeks, the patients can carry it out independently in their own homes or continue in in-hospital setting according to clinical condition and/or patient’s preference”.

4. The protocol lists several forms of training. Do you recommend that patients should do all of them or is there any criterion system for choosing which form of exercise is recommended for whom?

We agree that tables reporting protocol lists of several forms of training are excessive and confusing.

We change Table 1 deleting Traditional Chinese Medicine (not widely known, not strongly validated and not easy to propose) and reorganizing the scheme of exercise training based on the available data for COVID-19 patients at the moment in light of new Expert Consensus/Randomized Clinical Trial.

Moreover, we built Table 2 reporting COVID-19 different pathophysiology and related diseases.

Finally we built Table 3 reporting different exercise programs with general description, COVID-19 related disease validation data and specific COVID-19 data.

We consequently deleted table 2-6 making the paper much more easy reading.

We change the text accordingly.

5. Tables are difficult to read and the abbreviations used are inconsistent. Please standardise the abbreviations.
We changed the tables and we standardized abbreviations as suggested.

6. Traditional Chinese medicine respiratory rehabilitation is included in the protocol but there is no description or reference provided, which would be necessary, as this method is not widely known.

We agree that Traditional Chinese Medicine is not widely known, not strongly validated and not easy to propose. We preferred to delete it.

7. The leading symptom in many post-covid patients is a high heart rate at rest or on exercise. This problem is not mentioned in the article. What rehabilitation training is recommended for these patients?

We thank the reviewer for the interesting suggestion. We added “high heart-rate at rest and on exercise” among typical characteristic of COVID-19 patients in Table 2 with references about inappropriate sinus tachycardia in post COVID-19 patients tachycardia syndrome (American Autonomic Society Statement). Moreover, we consider cardiac and pulmonary patients as related pertinent disease for this feature and translate to COVID-19 patients the validated exercise program for these latter patients.

Round 2

Reviewer 1 Report

approved

Reviewer 2 Report

The authors have made the requested changes. I think that the manuscript has improved significantely.